# Big Data in Studying Acute Pain and Regional Anesthesia

**DOI:** 10.3390/jcm10071425

**Published:** 2021-04-01

**Authors:** Lukas M. Müller-Wirtz, Thomas Volk

**Affiliations:** 1Department of Anaesthesiology, Intensive Care and Pain Therapy, Saarland University Medical Center and Saarland University Faculty of Medicine, 66421 Homburg, Saarland, Germany; 2Outcomes Research Consortium, Cleveland, OH 44195, USA

**Keywords:** anesthesia, anesthesiology, big data, registries, database research, acute pain, pain management, postoperative pain, regional anesthesia, regional analgesia

## Abstract

The digital transformation of healthcare is advancing, leading to an increasing availability of clinical data for research. Perioperative big data initiatives were established to monitor treatment quality and benchmark outcomes. However, big data analyses have long exceeded the status of pure quality surveillance instruments. Large retrospective studies nowadays often represent the first approach to new questions in clinical research and pave the way for more expensive and resource intensive prospective trials. As a consequence, the utilization of big data in acute pain and regional anesthesia research has considerably increased over the last decade. Multicentric clinical registries and administrative databases (e.g., healthcare claims databases) have collected millions of cases until today, on which basis several important research questions were approached. In acute pain research, big data was used to assess postoperative pain outcomes, opioid utilization, and the efficiency of multimodal pain management strategies. In regional anesthesia, adverse events and potential benefits of regional anesthesia on postoperative morbidity and mortality were evaluated. This article provides a narrative review on the growing importance of big data for research in acute postoperative pain and regional anesthesia.

## 1. Introduction

This article provides a narrative review on the growing importance of big data for research in acute pain and regional anesthesia. We present an overview on big data initiatives and discuss further steps that are needed for a broader application of big data analytics. Finally, we present exemplary study results, which are derived from these databases and discuss future directions.

A randomized controlled trial is in principle considered the gold standard when best treatments are sought, because they are considered to have low risk of bias and confounding. However, a randomized controlled trial typically includes a fraction of patients for which generalizability may be poor. Furthermore, rare events are typically not used as the primary outcome due to the immense financial and logistical effort involved in including a large number of cases. In recent years, the concept of “real world data” has matured to complement results from randomized controlled trials studying treatments and exposures in more heterogenous populations recorded in large clinical databases, often referred to as “big data” [1].

Big data can be defined as “Extremely large datasets to be analyzed computationally to reveal patterns, trends, and associations, especially relating to human behavior and interactions.” [2]. In general, big data refers to databases with millions of data entries requiring advanced statistical methods and exceptional computing power. A potential downside of big data was recently exposed by the Facebook–Cambridge Analytica data scandal, when about 50 million Facebook profiles were harvested to analyze voters for the Trump election campaign [3]. However, with the intention to study medical outcomes and under the protection of data privacy by rigorous anonymization, large-scale databases represent a useful approach to extend current medical knowledge and aide decision making in health policy.

The rising importance of big data in anesthesiology and acute pain research is reflected by an exponential growth of PubMed search results related to the terms “big data” and “anesthesiology”, or “big data” and “pain” (Figure 1). This trend in medical research is underlined by several national initiatives to make big data accessible for research purposes. The British initiative NHS digital was launched in 2013 to use clinical data for nationwide research projects [4]. More recently, the Medical Informatics Initiative Germany was created in 2016 to facilitate exchange of medical data among German university hospitals for research [5]. A national anesthesia-specific big data initiative is the Danish Anaesthesia Database. It collects data on anesthesia-specific quality indicators and complications of about 70% of the Danish hospitals [6]. These exemplary initiatives underline how the digital transformation of healthcare systems will make large amounts of clinical data accessible for research in the upcoming years.

To make use of the increasing amount of data, clinical scientists are required to learn how to handle and analyze large datasets. Advanced statistical methods represent a cornerstone to produce robust results. Methods, such as multivariable regression [7,8,9] or propensity score matching [10], allow correction for confounders in order to isolate treatment or exposure effects. Regression methods may even distinguish direct and indirect (mediated) effects of exposures or treatments [11,12]. Finally, generalized estimating equations (GEE) or mixed effects models (MEM) are able to handle longitudinal data as often generated during anesthesia and perioperative medicine [13,14]. (Readers that are interested in these methods are directed to the provided references, which are short priming tutorials easy to understand even for non-statisticians.).

Another aspect that comes with big data is artificial intelligence and, specifically, machine-learning methods [15]. In traditional statistical models, predictors of an outcome, potential confounders, or mediators are actively defined and tested by the investigator based on biological plausibility and previous evidence. In contrast, machine learning in very general terms starts with the definition of an overall task including available input and the desired output variables. The model is trained with example data, automatically adapts to increase its predictive performance, and finally, is evaluated by applying it to data not included in the training process [15]. These models may help clinicians with additional information on diagnosis, treatment, risk, prognosis, and much more for a currently treated patient with several possible applications in anesthesiology [16,17]. Although this offers exceptional opportunities to improve medical care in the future, there are still obstacles to overcome before a broad application is feasible.

To evaluate the usefulness of big data analyses and artificial intelligence methods in acute pain and regional anesthesia research, we performed a literature search on PubMed using the terms “acute postoperative pain”, “opioids”, “regional anesthesia”, combined with the terms “database”, “registry”, “machine learning”, or the titles of the registries and databases given in Table 1. Exemplary studies for each source of big data were selected based on their population size and the clinical significance of their results.

## 2. Making Data Accessible for Research

In general, there are two approaches to make big data accessible for research purposes. The first represents the collection of data within clinical registries according to standardized protocols ensuring a high grade of validity. Another approach and most probably a game changer in clinical research is the unification of data standards for electronic health records and the use of detailed medical coding systems to facilitate data exchange and analyses across medical institutions.

The largest amount of clinical data is generated during the day-to-day clinical routine documentation and captured in a more or less structured way. Documentation practice markedly differs among healthcare systems, medical facilities, and individuals. However, the structure of the data is of outmost importance, because it may enhance or even totally exclude the accessibility for an analysis.

To enhance data quality, several clinical registries were created capturing research data by standardized protocols (Table 1). Examples for acute pain registries are the Improvement in Postoperative Pain Outcome Registry (PAIN OUT) [18,19] or the German equivalent initiative, the Quality Improvement in Postoperative Pain Management Registry (QUIPS) [20,21]. For regional anesthesia research, several registries with slightly different aims exist. One example is the German Network for Safety in Regional Anesthesia and Acute Pain Medicine registry (net-ra) [22,23] with recent studies focusing on potential complications of regional anesthesia procedures, such as infection, bleeding, or nerve injury.

Another approach to collecting clinical data for research purposes is the use of administrative databases (Table 1). For example, the commercial databases Premier [24,25] and MarketScan [26,27], which were primarily intended to support data-driven decision making in hospital management and health policy, provided data for several important large-scale studies. It may be naive to assume that administrative databases need no validation in the first place. Although the number of studies that use administrative data increases, there are only a few reports on data validity [28]. However, the increasing implementation of full electronic health records in these databases offers more options for data validation, which may help to solve previous issues with data validity.

While administrative data are often generated for billing purposes anyway, clinical registries have to be actively filled and supervised coming with an extra need for administrative staff or increased workload for medical staff. Automated data extraction and transfer is, thus, desirable and may additionally reduce potential transmission errors. Therefore, electronic health records should be collected in a digitally standardized format and then saved according to uniform data standards to ensure interoperability. For example, the Medical Informatics Initiative Germany created a core dataset with several modules based on the Fast Health Interop Resources (FHIR) [29] standard to facilitate the exchange of medical records among German university hospitals [5]. The COVID-19 pandemic has further fostered the implementation of uniform datasets. For example, the German Corona Consensus Dataset (GECCO) was recently created to enhance comparability of COVID-19 research data across studies and institutions [30].

Important core pieces of datasets are standardized coding systems, such as the International Classification of Diseases (ICD) [31] or the Systematized Nomenclature of Medicine Clinical Terms (SNOMED CT) [32] to code diagnoses, treatments, and outcomes. ICD represents the most widely used terminology, as it is often used for billing purposes. However, SNOMED CT is more versatile and detailed from a medical and scientific perspective [33]. The recognition of clinical terms and translation into data accessible for statistical analyses enables rapid extraction of data from electronic health records for large-scale retrospective cohort or case-control studies.

## 3. Big Data Initiatives in Acute Pain and Regional Anesthesia Research

Acute pain is not only a matter of discomfort. Poorly controlled postoperative pain is associated with impaired physical function, recovery, quality of life, and may even translate to persistent pain [34,35]. Despite these findings, about half of surgical patients complain about moderate to severe pain on the first day after surgery [36]. About 12% of surgical patients at one year after surgery continue to suffer from moderate to severe pain [37]. Regional anesthesia and analgesia techniques are important tools for postoperative pain management and may help to prevent negative consequences of acute postoperative pain [38,39]. In the following, we present some exemplary studies from the clinical registries and administrative databases presented in Table 1 that contributed to a better understanding of acute pain and regional anesthesia.

The Improvement in Postoperative Pain Outcome (PAIN OUT) registry is the first international acute pain registry and collects pain profiles on the first postoperative day [18,40]. It was established in 2009 funded by the European Union and merged with its German national counterpart, the Quality Improvement in Postoperative Pain Management (QUIPS) registry, in 2014 increasing its size to more than 550,000 datasets and more than 200 participating hospitals worldwide. One major aim of this initiative is to monitor quality of postoperative pain management to enable data-driven clinical decision making. A good example for this is an analysis including 50,523 patients from the QUIPS registry, which compared pain intensity on the first day after surgery among several surgical procedures. Most interestingly, particularly high pain scores were observed in procedures usually considered as minor, most probably because their analgesic needs are frequently underestimated [41]. A very recent study based on the PAIN OUT registry identified risk factors for poor postoperative pain outcomes in 50,005 patients. Risk factors were female sex, persistent pain, preoperative opioid intake, and young age. Combined to a simplified risk score they provided a prediction efficiency with an area under the receiver operating characteristics curve (AUC–ROC) of 0.7 (95%CI: 0.695–0.713) [15]. A further aim of the PAIN OUT registry is to evaluate the efficiency of analgesic treatments. Thus, the efficiency of epidural versus systemic analgesia for postoperative pain control was assessed in 2127 patients undergoing abdominal surgery. Epidural analgesia was associated with a lower risk for the perception of worst pain (RR: 0.75, 95%CI: 0.64–0.87) and time in severe pain (RR: 0.61, 95%CI: 0.5–0.75) [42]. Moreover, the influence of regional versus general anesthesia on postoperative pain and opioid consumption was determined in two cohorts of 2346 patients undergoing total knee replacement and 2315 patients undergoing total hip replacement. Regional anesthesia was associated with reduced opioid consumption (knee, OR: 0.20, 95%CI: 0.13–0.30; hip, OR: 0.17, 95%CI: 0.11–0.26), and in case of total knee replacements also with less pain (OR: 0.53, 95%CI: 0.36–0.78) [43]. In consistence, pain scores and patients’ satisfaction were improved in 15,326 patients undergoing knee replacement when regional anesthesia techniques were applied [44].

The Network for Safety in Regional Anesthesia and Acute Pain Medicine (net-ra) registry was established in 2007 by the German Society for Anesthesiology and Intensive Care Medicine (DGAI) and the Professional Association of German Anaesthetists (BDA) under support of the German Research Foundation (DFG). Preoperative, intraoperative, and postoperative data are collected from treating physicians at 25 German centers by a standardized form [23], and the registry currently includes more than 800,000 data entries. First use cases focused on studying adverse events in regional anesthesia. The net-ra investigators reported an incidence of spinal hematomas after epidural puncture of 0.15 per thousand based on 33,142 cases [45]. Among 8781 cases of the net-ra registry regional analgesia catheter-related infections (including light, moderate, and severe grades) were shown to be frequent with incidences of 2.7% for neuraxial and 1.3% for peripheral procedures [46]. As a consequence, many of the following net-ra studies focused on the identification of risk factors for catheter-related infections. For example, odds for catheter-related infections are increased in diabetics (OR: 1.26, 95%CI: 1.02–1.55) [47] or in obese patients with peripheral catheters (OR: 1.69, 95%CI: 1.25–2.28) [48]. Moreover, the catheter-related infection profile over up to 15 days was determined in a cohort of 44,555 patients showing that infection risk starts to increase slowly at the fourth day after catheter placement [49]. Potentially preventive treatments, such as the tunneling of thoracic epidural catheters and single-dose antibiotic prophylaxis were shown to be associated with about halving the odds for catheter-related infections [50,51]. Finally, benefits of ultrasound, nerve stimulation, and their combination for guiding peripheral nerve blocks were evaluated in 26,733 cases. The use of ultrasound alone reduced the odds of vascular and multiple skin punctures but increased the odds of paresthesia compared to the combined use of ultrasound and nerve stimulation [52].

The Pediatric Regional Anesthesia Network (PRAN) registry was established in 2007 to study regional anesthesia in children. With 21 participating centers, data from more than 100,000 regional anesthesia procedures were collected until today. The PRAN investigators reported a safety profile of regional anesthesia in children comparable to that in adults. In 104,393 blocks, no permanent neurologic deficit was reported. Incidences were 0.24 per thousand for transient neurologic deficits, 0.076 per thousand for local systemic toxicity, and 0.5% for cutaneous infections. There was only one epidural hematoma related to a bilateral paravertebral block and one epidural abscess needing surgical intervention [53]. The PRAN investigators further verified the safety of transversus abdominis plane blocks in children and neuraxial catheters in neonates [54,55]. A five to ten-fold spread in local anesthetic dose variability was observed in 40,121 peripheral nerve blocks in children underlining the lack of dose finding studies in children. In addition, wide variation in the administered local anesthetic dose and volume for caudal blocks in children ≤1 year was reported based on 14,367 cases from the PRAN registry, even though advantages of dose variation over a standardized dose are not supported by current evidence [56]. These findings support the development of practice guidelines to reduce unwarranted variability in local anesthetic dosing in pediatric regional anesthesia.

The IRORA emerged from the Australian and New Zealand Registry of Regional Anaesthesia (AURORA), which was established in 2006 to study the effectiveness and risks of peripheral nerve blocks. The IRORA investigators reported incidences of nerve injury of 0.4 per thousand and systemic local anesthetic toxicity of about 1 per thousand based on the first 8189 collected cases [57]. Consistent with low complication rates and high efficiency for acute postoperative pain control, almost 95% of 9969 patients from the IRORA registry were willing to undergo another peripheral nerve block procedure in the future [58]. The usefulness of registries to benchmark clinical treatment was impressively shown by Sites et al. when they boldly reported significantly higher pain scores and opioid use, as well as delayed discharge in their own hospital compared to most other IRORA member institutions [59]. Most recently, large variations in the local anesthetic dosing within and among hospitals were reported based on 26,457 peripheral blocks from the IRORA registry, which indicates again that dose finding studies and dosing guidelines are needed to reduce unwarranted variation in local anesthetic dosing [60].

The Anesthesia Quality Institute (AQI) was founded in 2008 by the American Society of Anesthesiologists and established the National Anesthesia Clinical Outcomes Registry (NACOR) [61]. NACOR collects perioperative data and helps participating institutions with providing quality reports and benchmarking to guide quality improvement efforts. In addition, it provides data for research purposes, but only a few investigations in the area of acute pain and regional anesthesia research have been conducted so far. Lam et al. used NACOR data to identify trends in the use of peripheral nerve blocks in mastectomy and lumpectomy procedures. In 189,854 surgical cases, the proportion of mastectomies performed with a peripheral nerve block gradually increased from 2010 to 2018 but still remained low with only 13% in 2018 [62]. The analysis of 108,625 patients receiving a total knee arthroplasty revealed that the use of regional anesthesia techniques increases with higher patient age, higher American Society of Anesthesiology scores, and when the anesthetist is board certified [63].

The Multicenter Perioperative Outcomes Group (MPOG) registry was established in 2008 at the US University of Michigan Medical School currently includes data on more than 15 million cases from 51 medical sites [64]. One of the first MPOG studies evaluated risk and outcomes of epidural hematomas after epidural catheter placements in 79,837 obstetric and 62,450 surgical patients. Notably, none of the obstetric patients but seven patients undergoing perioperative epidural catherization developed epidural hematomas that required surgical evacuation, suggesting a considerably lower risk for epidural bleeding complications in obstetric patients [65]. The risk for epidural hematomas after neuraxial regional anesthesia techniques was further evaluated in 1524 thrombocytopenic parturients. Risk for epidural hematoma increased considerably when platelet counts were below 100,000 mm^−3^, up to a potential risk of 11% when platelet counts were 0 to 49,000 mm^−3^ [66]. A currently ongoing MPOG project uses the registry’s infrastructure for a prospective collection of data on several patient- and procedure-specific factors on post-discharge pain profiles and opioid consumption patterns [67]. This project may help to complement the overall picture of the relationship between surgery and postoperative persistent pain.

The National Surgical Quality Improvement Program (NSQIP) registry emerged from the US National Veterans’ Affairs (VA) Surgical Quality Improvement Program (SQIP), which was established in 1991 to monitor surgical treatment quality in VA hospitals. In 2001, the American College of Surgeons (ACS) adopted the program and successfully extended its use to private hospitals [68]. Until today, there are more than 700 hospitals from 11 countries participating in NSQIP. Several studies used NSQIP data to evaluate the effect of regional anesthesia versus general anesthesia on postoperative outcomes. Saied et al. compared postoperative morbidity and 30-day mortality between 264,421 surgical patients receiving general anesthesia and 64,119 patients receiving regional anesthesia extracted from about 1.7 million surgical cases in the NSQIP registry. Regional anesthesia decreased length of stay and reduced the odds for several intraoperative and respiratory complications, or for experiencing any complication. However, there was no effect of anesthesia type on mortality [69]. Consistently, several smaller NSQIP studies including surgical subgroups reported no effect of regional anesthesia on mortality, but a clear reduction in postoperative complications by the use of regional anesthesia techniques [69,70,71,72,73,74,75,76,77].

US Medicare healthcare claims data were used to evaluate the influence of regional anesthesia on postoperative mortality and morbidity. The use of postoperative epidural analgesia was associated with a considerable reduction in 7-day (OR: 0.52, 95%CI: 0.38–0.73) and 30-day mortality (OR: 0.74, 95%CI: 0.63–0.88) in 68,723 Medicare patients [78]. Similar results were shown in Medicare patients undergoing colectomy [79], esophagectomy [80], and lung resection [81], but not in cases of total hip arthroplasties [82].

The Premier healthcare database contains data of more than 231 million patients from over 1000 US hospitals [24]. Until today, several large retrospective studies were conducted based on the Premier healthcare database. In the area of acute pain research, opioid-related side effects and multimodal pain management strategies were evaluated. Among 592,127 inpatient stays, opioid-induced respiratory depression was frequent in a range from 3% in obstetric to 17% in cardiothoracic/vascular surgery and was associated with a considerable increase in costs and length of stay [83]. A potential strategy to reduce opioid-related side effects is multimodal analgesia. In 181,182 obstructive sleep apnea patients, more than two additional non-opioid modes reduced the odds for mechanical ventilation (OR: 0.23, 95%CI: 0.16–0.32) and critical care admission (OR: 0.60, 95%CI: 0.48–0.75) [84]. In 512,393 hip and 1,028,069 knee arthroplasties, patients receiving more than two analgesic modes had fewer respiratory and gastrointestinal complications, as well as less opioid consumption and a shorter length of stay [85]. In 265,538 patients undergoing posterior lumbar fusion, the addition of NSAIDs/COX-2 inhibitors to opioids reduced opioid prescription, cost, and length of stay [86]. In 591,865 hip and 1,139,616 knee arthroplasties, the additional use of COX2-inhibitors (OR: 0.72, 95%CI: 0.67–0.76; OR: 0.82 95%CI: 0.78–0.86) and NSAIDs (OR 0.81, 95%CI: 0.77–0.85; OR: 0.90, 95%CI: 0.86–0.94) was associated with decreased odds for gastrointestinal complications [87]. In 190,057 patients older than 60 years with hip fractures, the use of NSAIDs reduced opioid consumption, length of stay, intensive care unit admissions, and pulmonary complications [88]. The additional use of acetaminophen likewise reduced opioid consumption and postoperative complications in 1,039,647 patients undergoing total hip or knee arthroplasty [89].

In the area of regional anesthesia, research data from the Premier healthcare database were used to evaluate potential benefits of regional anesthesia on postoperative morbidity and mortality. Memtsoudis et al. analyzed the data of 382,236 orthopedic patients and reported that general compared to neuraxial or combined neuraxial-general anesthesia was associated with increased 30-day mortality (OR: 1.83, 95%CI: 1.08–3.1; OR: 1.70, 95%CI: 1.06–2.74) [39]. Another study by Memtsoudis et al. with 795,135 patients undergoing total knee or hip replacement added that particularly older patients may benefit from receiving regional anesthesia due to a reduction in major complications [90]. A potential reduction in mortality by the use of regional anesthesia in orthopedic patients remains unclear, as this is in contrast to the above presented studies based on NSQIP and Medicare data. However, all presented studies consistently report a considerable reduction in postoperative complications by the utilization of regional anesthesia techniques.

The IBM MarketScan database was established in 1989 and collects healthcare claims data for privately and publicly insured people in the US for commercial and clinical analytics. Today, the database includes more than 237 million cases and was used to conduct the probably largest retrospective study on chronic opioid use. Sun et al. assessed risk factors for chronic opioid use in 641,941 opioid naive surgical patients and 18,011,137 opioid naive non-surgical patients. Especially patients undergoing orthopedic procedures, such as total knee arthroplasty (OR: 5.1, 95%CI: 4.7–5.6) and total hip arthroplasty (OR: 2.5, 95%CI: 2.1–3.0), next to open cholecystectomy (OR: 3.6, 95%CI: 2.8–4.6) and mastectomies (OR: 2.7, 95%CI: 2.3–3.1) had the highest odds for chronic opioid use [91]. The healthcare burden of continued postoperative opioid use was analyzed in 1,174,905 opioid-naive patients with an inpatient surgery and 2,930,216 opioid-naive patients with an outpatient surgery. An outpatient opioid prescription within 1 year after surgery was associated with considerably increased healthcare costs [92].

## 4. Artificial Intelligence and Machine-Learning Methods

The above presented examples of retrospective studies all used traditional statistical methods, which means that the predicting and confounding factors are actively chosen based on biological plausibility and previous clinical evidence. In contrast, machine-learning methods handle a variety of input data and may potentially identify important linkages that were previously unknown.

Machine-learning methods could be especially useful to predict acute postoperative pain. One of the earliest applications of machine-learning algorithms in pain research predicted moderate-to-severe postoperative pain with an accuracy of an AUC–ROC of 0.7 under usage of 796 clinical variables from 8071 surgical patients [93]. Another study used machine-learning models to predict persistent pain after breast cancer surgery. The final algorithm showed only a moderate sensitivity (80%) and specificity (70%), but patients not developing persistent pain were identified with an accuracy of 95% [94]. Moreover, machine-learning models predicted chronic opioid use after anterior cervical discectomy and fusion with an AUC–ROC of 0.8, and after knee arthroscopy with an AUC–ROC of 0.74. Finally, artificial intelligence may help clinicians with the prediction of postoperative opioid requirements in surgical patients [95].

To our knowledge, there is only one study that used machine-learning methods to study regional anesthesia on a population level. Tighe et al. used several machine-learning algorithms to predict the need for a femoral nerve block after anterior cruciate ligament repair [96]. The most accurate model was able to predict the need for a femoral nerve block with an AUC–ROC of 0.7, and the model was only trained with data from only 349 patients and only seven input variables [96]. We can only speculate to what extent accuracy may increase, once trained with millions of cases and a variety of input variables.

Compared to traditional statistical methods, an advantage is that machine-learning models trained with large datasets can be integrated into a patient data management system and repeatedly be retrained to enhance prediction accuracy and support clinical decision making. For example, a machine-learning model predicted clinical deterioration of patients on the ward more accurate than conventional regression methods [97], and a machine-learning model showed better performance in predicting intensive care unit readmissions than previously established clinical scores [98]. A deep-learning model predicted mortality in critically ill children with an AUC–ROC ranging from 0.89 to 0.97 six to 60 h before death [99]. Moreover, onset of sepsis was predictable 4–12 h before clinical recognition with an AUC–ROC ranging between 0.83 and 0.85 [100].

A potential source of error for artificial intelligence algorithms is that they rely on historical data. Data including wrong clinical decisions may be translated as potential solutions for future situations. Therefore, clinical data used to train algorithms need to be of high medical quality to obtain accurate predictions. Several studies are then needed to validate the algorithm. Despite great efforts, accuracy may still remain limited. For example, the IBM Watson for Oncology clinical decision-support system was frequently criticized for giving false recommendations [101,102]. Artificial intelligence methods should, thus, not be seen as a potential substitute for medical doctors, but as an additional tool to improve the information base for clinical decision making.

## 5. A Look into the Future

Analysis of big data is helpful to predict postoperative pain, the demand for pain medication, and for the evaluation of the effectiveness of multimodal pain treatment strategies. Future analyses should focus on evaluating negative consequences of acute postoperative pain on outcomes. As long as pain is regarded as an accompanying phenomenon of surgery, and not as a potentially life-threatening condition, acute postoperative pain will continue to be undertreated. We are convinced that future big data analyses will further clarify the negative consequences of acute postoperative pain, which should finally be regarded as a central quality indicator of perioperative care.

Large-scale retrospective studies strongly suggest improved outcomes with regional anesthesia in surgical patients. As a consequence, there are reasonably large trials running that compare the outcome of spinal versus general anesthesia in hip fracture patients, such as the REGAIN trial or its German sister trial iHOPE [103,104]. Future big data analyses may focus on identifying which patients and surgical procedures benefit most from regional anesthesia to complement the overall picture and pave the way for further randomized controlled trials.

Statistical methods currently applied to deal with big data could be challenged in the future. The most common approaches are the application of multiple regression methods or propensity score matching followed by regression to adjust for confounding. Artificial intelligence algorithms, in particular machine-learning methods, may be useful for outcome and risk prediction. There are currently only a few studies in the field of acute postoperative pain and regional anesthesia research using machine-learning methods, but there may be great potential of these methods when trained with big data.

## 6. Conclusions

Big data has repeatedly been shown to be quite useful for studying acute pain and regional anesthesia. Most perioperative big data initiatives were established to monitor treatment quality and benchmark outcomes. However, big data analyses have long exceeded the status of pure quality surveillance instruments. Large retrospective studies nowadays often represent the first approach to new questions in clinical research and may pave the way for expensive and resource intensive prospective investigations. Of note, despite the excitement for big data, these studies remain retrospective and reveal a number of potential biases. Even the largest retrospective study using the most sophisticated statistical methods is only as accurate as the confounders that are known, measured, and adjusted for. Randomization is still the most powerful tool to eliminate most bias. Therefore, randomized controlled trials are still providing the highest grade of clinical evidence and several of them must follow and validate the results of retrospective studies. However, big data studies provide important insights into “real-world” data to complement the overall picture of diseases, outcomes, and potential treatments. Artificial intelligence may further help to increase accuracy of big data analysis, but the advantage over traditional statistical methods in population-based research still needs to be proven. For now, a reasonable clinical use of these new methods in pain medicine could be the prediction of postoperative pain outcomes alerting the clinician to focus treatment efforts on specific patients at risk. Overall, we believe that big data is a useful approach to study acute pain and regional anesthesia, and that it helps to improve perioperative care.

## Figures and Tables

**Figure 1 jcm-10-01425-f001:**
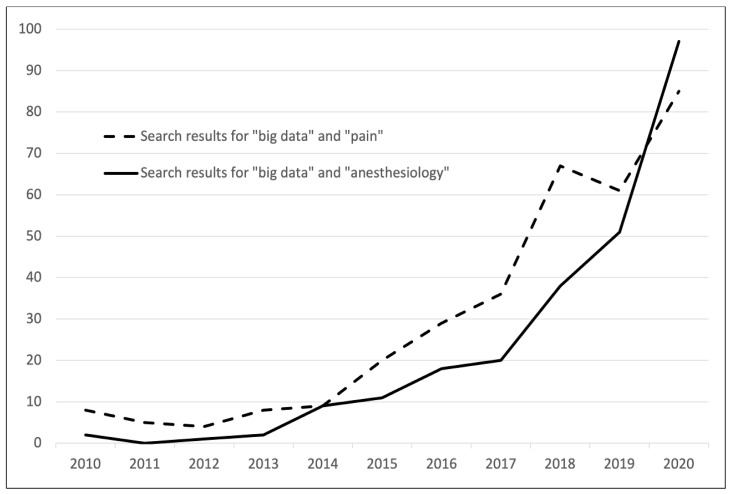
Search results on PubMed for “big data” and “anesthesiology” or “pain” from 2010 to 2020.

**Table 1 jcm-10-01425-t001:** Sources of big data for acute pain and regional anesthesia research.

Short Title	Title	Est.	Webpage
Acute postoperative pain on the first postoperative day			
PAIN OUT	Improvement in Postoperative Pain Outcome	2009	pain-out.med.uni-jena.de
QUIPS	Quality Improvement in Postoperative Pain Management	2005	quips-projekt.de
Regional anesthesia and acute postoperative pain			
net-ra	German Network for Safety in Regional Anesthesia and Acute Pain Medicine	2007	net-ra.eu
Regional anesthesia			
PRAN	Pediatric Regional Anesthesia Network	2007	pedsanesthesia.org
IRORA	International Registry of Regional Anesthesia	2006	regionalanaesthesia.wordpress.com
Administrative databases			
Medicare	Medicare and Medicaid healthcare claims database	1999	medicare.gov resdac.org
Premier	Premier healthcare database	1997	premierinc.com
MarketScan	IBM MarketScan research database (previously: Truven Health MarketScan Database)	1989	ibm.com/products/marketscan-research-databases
Anesthesiology and Perioperative Medicine			
NACOR	National Anesthesia Clinical Outcomes Registry	2008	aqihq.org
MPOG	Multicenter Perioperative Outcomes Group	2008	mpog.org
NSQIP	American College of Surgeons National Surgical Quality Improvement Program	1991	facs.org

All webpages presented in the table were last accessed on 22 February 2021.

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
