# Peer review of "Big Data in Studying Acute Pain and Regional Anesthesia"

_jcm, 2021, doi:10.3390/jcm10071425_

Round 1

Reviewer 1 Report

Review: Big data in studying acute pain and regional anesthesia

This is an extensive review of the use of big data sets and analysis in the field of acute pain and regional anesthesia. The authors should be commended for their detailed evaluation of the currently literature and their well written article. I have no major concerns but I would offer suggestions only to hopefully, improve readability.

Introduction: The last paragraph really should be moved to the beginning of this section or a shorter version of the same information. It would help the reader to know where the authors intend to direct the manuscript. It is nicely summarized but it feels that the introduction section could be focused with a topic paragraph.

Page 2 Line 49 “foster the implication of digital health care services [4].” It isn’t clear to me what the authors meant by this phrase. Could they express their idea with a different word or more detail.

Making data accessible for research: Again, it would be helpful if there was at least a sentence or two as introduction to this section. The last paragraph serves this but I felt that I read for a page before I knew what they were driving at.

Page 3 line 88. The paragraph is started with Therefore but it isn’t clear what key idea of the paragraph above the idea is being referred to. Can Therefore be replaced with the idea that led to the development of clinical registries the authors were trying to convey.

Page 3 line 96 “Another approach” again, starting a paragraph with this leads the reader to question “Another approach to what?” Can the authors identify what idea they were trying to carry through?

Page 3 line 100 In the phrase “Although the amount of investigations increases,” Are the authors referring to querying databases or the construction of databases?

Big data initiatives in acute pain and regional anesthesia research

Page 6 line 166 and the rest of this paragraph. Is the Network registry net-ra only seen as lower case by that organization? I could not find this specifically on the internet. If it could be referred to as NET-RA it will be easier for the reader to follow this through the paragraph.

Page 7 Line 197 the number 40 121 is missing a comma

Page 7 line 207 the number 8189 is missing a comma

Page 7 line 220 I addition is a typo and should be In

Page 7 line 222 “research were conducted.” Might reader better as “research have been conducted.”

Artificial intelligence and machine learning methods

Page 9 line 325 “extend” likely is a typo and should be “extent”

A look into the future

It might be better to start the paragraph as “Analysis of big data…” It isn’t the big data per se that you are referring to here.

Reviewer 2 Report

In the manuscript entitled “Big data in studying acute pain and regional anesthesia” the authors performed a comprehensive narrative review of the use and usefulness of big data and artificial intelligence in the fields of postoperative pain and regional anesthesia, summarising the current state of research on this topic.

The article main subject, big data collection and analysis, represents a major trend in modern medicine. It also stands as essential tool for anaesthetic management and perioperative care improvement.

The review is well written and logically structured in six parts including an introduction, an overview followed by a detailed presentation of the main research resources - clinical registries and administrative databases, the role of algorithm-derived machine learning methods and artificial intelligence in the prediction of the acute and chronic postoperative pain and the need for regional anesthesia, future perspectives and conclusions. 

Minor issue

At the moment, artificial intelligence and machine learning methods, have several drawbacks that need to be underlined. The authors already mentioned in lines 74-75, “Although this offers exceptional opportunities to improve medical care in the future, there are still obstacles to overcome before a broad application is feasible.” Please detail the actual obstacles and pitfalls, (e.g., potential iatrogenic risks associated with flawed medical algorithm as IBM Watson Health’s cancer algorithm);  the need for audit, simulation, and prospective validation before using in clinical practice.

Overall, I want to congratulate the authors for the language clarity, very good information flow and excellent scientific content.

Reviewer 3 Report

Thank you for requesting my participation in the review of the manuscript "Big data in studying acute pain and regional anesthesia".

The manuscript is well written, little confusing, and could be of interest to several readers. It is also "in tune" with the current trend towards big data research.

I would have two comments that could enhance the readability and the fluidity of the manuscript.

It would be interesting if the authors did write the manuscript in the form of a research study. Why did the authors choose this topic? What do they want to prove? How did they conduct their research to prove their point? What or how many articles did they analyze? Which databases did they use? etc.

It would also be interesting to subdivide the manuscript into an introduction and a research question, a research methodology, and then present the results and the discussion (these two parts are the entire current article).

I hope this can help.

Issam.

Round 2

Reviewer 3 Report

Thank you for this thorough and thoughtful review of the manuscript.

Issam